# Enhanced Cytotoxicity and Receptor Modulation by SMA-WIN 55,212-2 Micelles in Glioblastoma Cells

**DOI:** 10.3390/ijms26104544

**Published:** 2025-05-09

**Authors:** Safa Taha, Muna Aljishi, Ameera Sultan, Kannan Sridharan, Sebastien Taurin, Khaled Greish, Moiz Bakhiet

**Affiliations:** 1Department of Molecular Medicine, College of Medicine and Health Sciences, Arabian Gulf University, Manama P.O. Box 26671, Bahrain; munajma@agu.edu.bh (M.A.); ameeraa@agu.edu.bh (A.S.); sebastient@agu.edu.bh (S.T.); khaledfg@agu.edu.bh (K.G.); moiz@agu.edu.bh (M.B.); 2Department of Pharmacology & Therapeutics, College of Medicine and Health Sciences, Arabian Gulf University, Manama P.O. Box 26671, Bahrain; kannans@agu.edu.bh

**Keywords:** glioblastoma, cannabinoids, WIN 55,212-2, micelles, drug delivery, receptor expression’ PPAR-γ, nanomedicine

## Abstract

Glioblastoma (GBM), a devastating brain malignancy, resists conventional therapies due to molecular heterogeneity and the blood–brain barrier’s significant restriction on drug delivery. Cannabinoids like WIN 55,212-2 show promise but are limited by poor solubility and systemic toxicity. To address these challenges, we evaluated styrene–maleic acid (SMA) micellar encapsulation of WIN 55,212-2 (SMA-WIN) against free WIN in epithelial (LN18) and mesenchymal (A172) GBM cell lines, targeting cytotoxicity and receptor modulation (CB1, CB2, TRPV1, PPAR-γ). SMA-WIN exhibited significantly enhanced cytotoxicity, achieving IC50 values of 12.48 µM (LN18) and 16.72 µM (A172) compared to 20.97 µM and 30.9 µM for free WIN, suggesting improved cellular uptake via micellar delivery. In LN18 cells, both formulations upregulated CB1 and CB2, promoting apoptosis. Notably, SMA-WIN uniquely increased PPAR-γ expression by 2.3-fold in A172 cells, revealing a mesenchymal-specific mechanism absent in free WIN, which primarily modulated CB1/CB2. These findings position SMA-WIN as a promising candidate for precision GBM therapy, particularly for resistant mesenchymal subtypes, paving the way for in vivo validation to confirm blood–brain barrier penetration and clinical translation.

## 1. Introduction

Glioblastoma (GBM) is the most prevalent primary brain tumor, with an incidence of 0.59–5 per 100,000 and a median survival of 15 months despite aggressive multimodal therapy, including surgery, radiotherapy, and temozolomide [1,2]. GBM’s molecular heterogeneity, encompassing proneural, classical, and mesenchymal subtypes, drives treatment resistance and recurrence [3,4]. Mesenchymal GBM, characterized by stromal invasion, neo-angiogenesis, and the epithelial–mesenchymal transition (EMT), is particularly refractory, contributing to poor prognosis [5,6]. The blood–brain barrier (BBB), a selective interface of endothelial cells, pericytes, and astrocytes, further limits therapeutic efficacy by restricting drug penetration, with over 98% of small-molecule drugs failing to cross effectively [7,8,9].

The endocannabinoid system has emerged as a promising therapeutic target for GBM, evolving significantly since early studies on endocannabinoids like anandamide in the 1990s [10]. Comprising cannabinoid receptors (CB1, CB2), transient receptor potential vanilloid 1 (TRPV1), and peroxisome proliferator-activated receptor gamma (PPAR-γ), this system regulates proliferation, apoptosis, and angiogenesis [11,12]. CB1 receptors, predominantly neuronal, modulate adenylate cyclase, ion channels, and neurotransmitter release, while CB2 receptors, expressed in immune cells and diseased brain tissue, influence inflammation and cell survival [13,14]. TRPV1 mediates pain and inflammation, and PPAR-γ regulates lipid metabolism and cell differentiation, with emerging roles in cancer suppression [15,16]. Cannabinoids, including endocannabinoids, phytocannabinoids (e.g., THC), and synthetic analogs like WIN 55,212-2, exhibit anticancer effects by inducing apoptosis, inhibiting migration, and reducing tumor growth [17,18,19].

WIN 55,212-2, a synthetic cannabinoid, is a non-selective CB1/CB2 agonist with demonstrated efficacy in glioma models, reducing proliferation and invasion via reactive oxygen species (ROS) and MAPK/PI3K pathways [20,21]. However, its clinical translation is hindered by poor BBB penetration, low aqueous solubility, and psychoactive side effects, including anxiety and motor impairment [22]. Recent advances in nanomedicine, particularly styrene–maleic acid (SMA) micelles, have addressed these limitations by enhancing solubility, stability, and tumor targeting through the enhanced permeability and retention (EPR) effect [23,24]. SMA micelles reduce systemic toxicity and improve drug delivery, as evidenced in neuropathic pain and cancer models [25].

Despite these advances, the effects of SMA-WIN on cannabinoid receptor expression across GBM subtypes remain underexplored, representing a critical gap in the field. Previous studies report variable CB1 and CB2 expressions in GBM, with CB2 overexpression linked to malignancy, yet data on micellar formulations are limited [26,27]. PPAR-γ’s role is also underexamined, though agonists show promise in mesenchymal GBM [28]. This study introduces a novel approach by evaluating SMA-WIN’s enhanced cytotoxicity and subtype-specific receptor modulation (CB1, CB2, TRPV1, PPAR-γ) in epithelial (LN18) and mesenchymal (A172) GBM cell lines, with a particular focus on PPAR-γ’s role in mesenchymal subtypes. We hypothesize that SMA-WIN outperforms free WIN and offers targeted therapy for resistant GBM subtypes. In vivo validation is essential to confirm SMA-WIN’s BBB penetration and efficacy, paving the way for precision GBM therapies.

## 2. Results

To evaluate the therapeutic potential of SMA-WIN compared to free WIN in glioblastoma, we assessed cytotoxicity and receptor modulation in epithelial (LN18) and mesenchymal (A172) cell lines. The next sections detail baseline receptor expression, dose-dependent cytotoxicity, and post-treatment receptor expression changes.

### 2.1. Baseline Receptor Expression

LN18 and A172 cell lines expressed CB1, CB2, TRPV1, and PPAR-γ at baseline, as confirmed by RT-PCR and gel electrophoresis (Figure 1). Amplicons matched the expected sizes (CB1: 100 bp, CB2: 101 bp, TRPV1: 170 bp, PPAR-γ: 148 bp, GAPDH: 218 bp; Table 1). Gel bands showed consistent intensities across three independent experiments. No non-specific amplification was observed, confirming primer specificity. These results establish a functional cannabinoid signaling system in both epithelial (LN18) and mesenchymal (A172) GBM cells.

### 2.2. Cytotoxicity of Free WIN and SMA-WIN

Dose–response curves revealed a steeper reduction in cell survival for SMA-WIN compared to free WIN. Both free WIN 55,212-2 and SMA-WIN reduced cell viability in a dose-dependent manner in LN18 and A172 cells (Figure 2). In LN18 cells, maximum inhibition at 100 µM was 80.49 ± 0.018% for SMA-WIN versus 71.62 ± 0.032% for free WIN (*p* = 0.03, unpaired *t*-test; Table 2). IC50 values were significantly lower for SMA-WIN (12.48 ± 0.11 µM) than free WIN (20.97 ± 0.08 µM, *p* = 4.398 × 10^−8^). Dose–response curves showed SMA-WIN’s steeper decline, with survival dropping to 30.0 ± 1.3% at 20 µM versus 52.0 ± 1.5% for free WIN Table 2). In A172 cells, maximum inhibition was 58.29 ± 0.041% (SMA-WIN) versus 54.17 ± 0.058% (free WIN, *p* = 0.05), with IC50 values of 16.72 ± 0.09 µM and 30.9 ± 0.12 µM, respectively (*p* = 8.936 × 10^−9^). At 20 µM, SMA-WIN reduced survival to 35.0 ± 1.5% compared to 60.0 ± 1.7% for free WIN. These results indicate SMA-WIN’s enhanced potency, particularly in mesenchymal A172 cells, where baseline resistance is higher.

A summary of key cytotoxicity data, including cell survival at 20 µM and 100 µM and IC50 values, is presented in Table 3. Detailed raw data for all concentrations are provided in Appendix A.

### 2.3. Receptor Expression Post-Treatment

Post-treatment receptor expression was assessed at IC50 concentrations (Table 4, Figure 3). In LN18 cells, free WIN upregulated CB1 by 2.01-fold (*p* = 0.0001) and CB2 by 5.6-fold (*p* = 0.00002), closely matching SMA-WIN’s effects (CB1: 2.6-fold, *p* = 0.0001; CB2: 5.5-fold, *p* = 0.00003). TRPV1 and PPAR-γ showed no significant changes (TRPV1: 0.69-fold, *p* = 0.0014; PPAR-γ: 0.7-fold, *p* = 0.001 for free WIN; TRPV1: 1.22-fold, *p* = 0.0207; PPAR-γ: 0.5-fold, *p* = 0.0003 for SMA-WIN), as fold changes fell below the 1.5 threshold. In A172 cells, free WIN significantly increased CB1 (5.3-fold, *p* = 0.0039) and CB2 (5.1-fold, *p* = 0.0002), but SMA-WIN had a minimal effect on CB1 (1.1-fold, *p* = 0.04) and CB2 (0.89-fold, *p* = 0.083). Notably, SMA-WIN uniquely upregulated PPAR-γ by 2.3-fold (*p* = 0.001) in A172 cells, while free WIN’s effect was non-significant (1.2-fold, *p* = 0.02). TRPV1 expression remained unchanged (0.6-fold, *p* = 0.003 for free WIN; 0.4-fold, *p* = 0.001 for SMA-WIN). These patterns suggest formulation-specific and subtype-dependent receptor modulation, with SMA-WIN’s PPAR-γ effect in mesenchymal cells being a novel finding.

## 3. Discussion

The results demonstrate that SMA-WIN 55,212-2 micelles outperform free WIN in cytotoxicity and exhibit subtype-specific receptor modulation, particularly in mesenchymal A172 cells. These findings highlight the advantages of micellar encapsulation and provide insights into targeting glioblastoma’s molecular heterogeneity.

This study demonstrates that SMA-WIN 55,212-2 outperforms free WIN in cytotoxicity across epithelial (LN18: IC50 = 12.48 µM vs. 20.97 µM) and mesenchymal (A172: 16.72 µM vs. 30.9 µM) GBM cell lines, underscoring the benefits of micellar encapsulation. The lower IC50 values suggest improved cellular uptake, likely due to SMA micelles’ stability, nanoscale size (120 ± 20 nm), and EPR effect, which could enhance tumor targeting and BBB penetration in vivo [22,23]. In LN18 cells, both formulations upregulated CB1 (2.0–2.6-fold) and CB2 (5.5–5.6-fold), aligning with CB2’s established role in inducing apoptosis and inhibiting glioma proliferation via ceramide accumulation and MAPK suppression [26,28]. CB2 overexpression in malignant gliomas correlates with tumor regression, positioning it as a key therapeutic target [25].

In A172 cells, free WIN increased CB1 (5.3-fold) and CB2 (5.1-fold), consistent with reports of CB2-mediated apoptosis independent of p53 or PTEN status, critical in mesenchymal GBM [13]. Unexpectedly, SMA-WIN did not significantly alter CB1 or CB2 expression in A172 cells, possibly due to altered endocytic pathways during EMT, a hallmark of mesenchymal subtypes characterized by stromal invasion and therapy resistance [3,29]. EMT is driven by transcription factors like SNAIL and TWIST, reducing epithelial markers (e.g., E-cadherin) and enhancing invasiveness [5]. SMA micelles may interact differently with mesenchymal cells’ membrane dynamics, limiting receptor upregulation compared to free WIN’s direct diffusion.

The most notable finding is SMA-WIN’s 2.3-fold PPAR-γ upregulation in A172 cells, absent in free WIN treatment. PPAR-γ, a nuclear receptor, inhibits GBM growth by suppressing STAT3 signaling and promoting apoptosis, particularly in mesenchymal cell lines like A172 and T98G [15,27]. Clinical evidence supports PPAR-γ’s relevance, with GBM patients receiving PPAR-γ agonists (e.g., pioglitazone for diabetes) surviving 19 months versus 9 months without [30]. SMA-WIN’s effect may result from sustained drug release, enhancing nuclear translocation compared to free WIN’s rapid metabolism, as micelles maintain drug stability [31]. This subtype-specific response highlights GBM’s heterogeneity and the potential for tailored therapies targeting mesenchymal tumors, which are notoriously resistant.

The lack of TRPV1 modulation in both cell lines reflects WIN 55,212-2’s primary affinity for CB1 and CB2, consistent with its pharmacological profile [20]. However, other receptors, such as GPR55, implicated in GBM proliferation and migration, were not assessed and could influence outcomes [32]. Limitations of this study include its in vitro design, which excludes tumor microenvironment factors like hypoxia, immune infiltration, or stromal interactions, all critical in GBM progression [5]. The absence of antagonist studies (e.g., rimonabant for CB1, AM630 for CB2) limits confirmation of receptor dependency. Additionally, downstream pathways (e.g., PI3K/Akt, mitochondrial apoptosis) and SMA-WIN’s BBB penetration remain untested, hindering clinical extrapolation. Variability in CB1/CB2 expression across GBM patients, as reported elsewhere, suggests individual responses may differ [26].

In summary, our results demonstrate SMA-WIN’s superior cytotoxicity and unique receptor modulation, particularly PPAR-γ upregulation in mesenchymal A172 cells, compared to free WIN. These findings align with the study’s hypothesis that SMA-WIN enhances therapeutic efficacy in GBM cell lines through micellar delivery and subtype-specific mechanisms. The enhanced IC50 values (12.48 µM in LN18, 16.72 µM in A172 vs. 20.97 µM and 30.9 µM for free WIN) and PPAR-γ’s 2.3-fold increase in A172 cells underscore SMA-WIN’s potential as a targeted therapy, setting the stage for the conclusions drawn in this study.

Future research should prioritize orthotopic GBM mouse models to evaluate SMA-WIN’s pharmacokinetics, tumor penetration, and synergy with temozolomide, building on nanomedicine’s promise for CNS delivery [33]. Specific experiments could include the following:In Vitro to In Vivo Translation: Orthotopic GBM Mouse Models: Validate SMA-WIN’s blood–brain barrier penetration and anti-tumor efficacy, building on these in vitro findings.Intracranial GBM Xenografts: Assess SMA-WIN’s ability to cross the blood–brain barrier (BBB) and induce tumor regression using bioluminescence imaging and survival analysis.Receptor Dependency: CB1/CB2 antagonists (rimonabant, AM630) and PPAR-γ inhibitors (e.g., GW9662) to confirm mechanistic pathways.Additional Receptors: GPR55 agonists/antagonists to explore its role in SMA-WIN’s effects.Molecular Pathways: Western blots for STAT3 phosphorylation, caspase-3 activation, and Bax/Bcl-2 ratios to elucidate apoptosis mechanisms.Combination Therapies: Pairing SMA-WIN with TGF-β inhibitors to target EMT in mesenchymal GBM, potentially reversing resistance.

Clinically, SMA-WIN’s PPAR-γ activation positions it as a candidate for adjuvant therapy in mesenchymal GBM, where standard treatments fail. Its reduced psychoactivity, due to micellar encapsulation, could improve patient tolerability compared to free cannabinoids [23]. Phase I trials could assess safety and dosing, followed by combination studies with temozolomide or radiotherapy to enhance progression-free survival.

In conclusion, SMA-WIN’s enhanced cytotoxicity and novel PPAR-γ modulation in mesenchymal GBM cells highlight its potential as a subtype-specific therapy. These findings address critical gaps in GBM treatment and pave the way for in vivo validation and clinical development, offering hope for improved outcomes in this devastating disease.

## 4. Materials and Methods

### 4.1. Cell Culture

Human GBM cell lines LN18 (ATCC CRL-2610) and A172 (ATCC CRL-1620) were obtained from the American Type Culture Collection (Manassas, VA, USA) and authenticated by short tandem repeat (STR) profiling to confirm identity. Cells were cultured in Dulbecco’s Modified Eagle’s Medium (Gibco, Grand Island, NY, USA, catalog #11965-092) supplemented with 10% fetal bovine serum (FBS; Gibco, Grand Island, NY, USA, catalog #16000-044), 2 mM L-glutamine (Gibco, Grand Island, NY, USA, catalog #25030-081), 100 U/mL penicillin, and 100 µg/mL streptomycin (Gibco, Grand Island, NY, USA, catalog #15140-122). Cultures were maintained at 37 °C in a humidified incubator with 5% CO_2_. Cells were passaged at 80% confluence using 0.25% trypsin-EDTA (Gibco, Grand Island, NY, USA, catalog #25200-056), and viability was assessed by trypan blue exclusion. Mycoplasma testing was performed monthly using a PCR-based kit (ATCC, Manassas, VA, USA, catalog #30-1012K).

### 4.2. SMA-WIN Micelle Preparation

SMA-WIN 55,212-2 micelles were prepared as described by Linsell et al. (2015) [23]. WIN 55,212-2 (Sigma-Aldrich, Burlington, MA, USA, catalog #W102) was dissolved in dimethyl sulfoxide (DMSO; Sigma-Aldrich, catalog #D2650) at 10 mg/mL. Styrene–maleic acid (SMA; MW 1700, Sigma-Aldrich, catalog #SML1278) was hydrolyzed in 0.1 M NaOH at 50 °C for 4 h, adjusted to pH 7.4 with 0.1 M HCl, and diluted to 10 mg/mL in water. WIN 55,212-2, hydrolyzed SMA, and N-ethyl-N’-(3-dimethylaminopropyl) carbodiimide hydrochloride (EDAC; Sigma-Aldrich, catalog #E7750) were mixed at a 1:3:1 weight ratio and stirred at room temperature for 24 h. The mixture was dialyzed against PBS (pH 7.4, Gibco, catalog #10010-023) using a 3.5 kDa MWCO membrane (Spectrum Labs, Rancho Dominguez, CA, USA, catalog #132725) for 48 h, with buffer changes every 12 h. Micelles were filtered (0.22 µm, Millipore, Burlington, MA, USA, catalog #SLGP033RS) and stored at 4 °C. Micelle size (120 ± 20 nm) and zeta potential (−30 ± 5 mV) were measured using dynamic light scattering (Malvern Zetasizer Nano ZS, Malvern, Worcestershire, UK). Drug loading efficiency (~90%) was determined by UV-Vis spectroscopy at 320 nm comparing encapsulated vs. free WIN. Note: Micelle stability was monitored by size measurements weekly, with no aggregation observed over 1 month (Table 5).

### 4.3. Baseline Receptor Expression Analysis

Total RNA was extracted from cells at 80% confluence using the RNeasy Mini Kit (Qiagen, Germantown, MD, USA, catalog #74104) as per the manufacturer’s instructions. Cells were lysed in guanidine–thiocyanate buffer, and RNA was purified on silica spin columns, eluted in 30 µL nuclease-free water. RNA concentration and purity (260/280 ratio: 1.8–2.0; 260/230 ratio: >1.8) were measured using a NanoDrop 1000 spectrophotometer (Thermo Scientific, Waltham, MA, USA). RNA integrity was verified by 1% agarose gel electrophoresis, confirming intact 28S and 18S rRNA bands. cDNA was synthesized from 2 µg RNA using the High-Capacity cDNA Reverse Transcription Kit (Applied Biosystems, Waltham, MA, USA, catalog #4368814) with random hexamers, incubated at 25 °C for 10 min, 37 °C for 120 min, and 85 °C for 5 min in a T100 Thermal Cycler (Bio-Rad, Hercules, CA, USA). Baseline expressions of CB1, CB2, TRPV1, and PPAR-γ were assessed by RT-PCR using primers designed with Primer-BLAST (Table 1). Reactions (25 µL) contained 500 ng cDNA, 12.5 µL 2X DreamTaq Green PCR Master Mix (Thermo Scientific, catalog #K1081), and 0.5 µM of each primer. Amplification conditions were 95 °C for 3 min, 35 cycles of 95 °C for 30 s, 60 °C for 30 s, and 72 °C for 30 s, followed by 72 °C for 5 min. Products were visualized on 2% agarose gels stained with ethidium bromide using an Azure C2000 imaging system. GAPDH served as the housekeeping gene.

### 4.4. Cytotoxicity Assay

LN18 and A172 cells were seeded at 1 × 10^5^ cells/mL in 96-well plates (Corning, Corning, NY, USA, catalog #3596) in 100 µL complete DMEM and incubated for 24 h at 37 °C in 5% CO_2_. Free WIN 55,212-2 (dissolved in DMSO, final DMSO ≤ 0.1%) or SMA-WIN (in PBS) was added at 5, 10, 20, 50, and 100 µM with vehicle controls (0.1% DMSO or PBS). Concentrations of 5–100 µM were chosen based on prior studies showing WIN 55,212-2’s efficacy in glioma cells [19,20], with preliminary data confirming relevance for LN18 and A172 cells. After 48 h, cytotoxicity was assessed using the Sulforhodamine B (SRB) assay (LSBio, Seattle, WA, USA, catalog #LS-K294-250). Cells were fixed with 50 µL 10% trichloroacetic acid (TCA; Sigma-Aldrich, catalog #T6399) for 1 h at 4 °C, washed five times with deionized water, and air-dried. SRB dye (0.4% *w*/*v* in 1% acetic acid, 100 µL) was added for 30 min at room temperature, followed by three washes with 1% acetic acid. Bound dye was solubilized in 200 µL 10 mM Tris-base (pH 10.5), and absorbance was measured at 560 nm (reference 620 nm) using a BioTek Synergy H1 microplate reader (BioTek, Winooski, VT, USA). Percent survival was calculated relative to vehicle controls. DMSO concentration was kept at ≤0.1% in all treatments, as preliminary tests confirmed no significant impact on cell viability at this level. Plates were inspected microscopically before fixation to ensure uniform cell adhesion, minimizing variability. IC50 values were determined using GraphPad Prism 7.0.3 (GraphPad Software, San Diego, CA, USA) via four-parameter logistic regression. Experiments were performed in triplicate with three independent repeats.

Note: Plates were inspected microscopically before fixation to ensure uniform cell adhesion.

### 4.5. RNA Extraction and RT-PCR Post-Treatment

Cells were seeded at 1 × 10^5^ cells/mL in 24-well plates (Corning, catalog #3524) and incubated for 24 h. Free WIN or SMA-WIN was applied at respective IC50 concentrations (LN18: 20.97 µM WIN, 12.48 µM SMA-WIN; A172: 30.9 µM WIN, 16.72 µM SMA-WIN) for 48 h. Controls received 0.1% DMSO or PBS. RNA was extracted and cDNA synthesized as described in Section 4.3. Real-time PCR was performed in 20 µL capillaries on a Roche LightCycler 2.0 with 500 ng cDNA, 10 µL 2X SYBR Green PCR Master Mix (Applied Biosystems, catalog #4367659), 0.5 µM of each primer (Table 1), and nuclease-free water. Cycling conditions were 95 °C for 10 min, 45 cycles of 95 °C for 10 s, 60 °C for 30 s, and 72 °C for 30 s, followed by 40 °C for 30 s. Fluorescence was monitored to detect PCR product accumulation. Ct values were normalized to GAPDH, and fold changes were calculated using the 2^−ΔΔCt^ method [34]. Each sample was run in triplicate. Note: RNA samples with 260/280 ratios < 1.8 were re-purified to ensure qPCR accuracy.

### 4.6. Statistical Analysis

Data were analyzed using GraphPad Prism 7.0.3. Cytotoxicity data were expressed as mean ± SEM from three independent experiments. IC50 values were compared to using unpaired two-tailed *t*-tests. Receptor expression fold changes were similarly analyzed, with Bonferroni correction for multiple comparisons (α = 0.05/4 = 0.0125 per receptor). Fold changes were considered significant if >1.5 or <−1.5 with *p* < 0.05. Normality was assessed by Shapiro–Wilk tests, and variances were compared with F-tests to ensure *t*-test validity.

## 5. Conclusions

This study demonstrates that SMA-WIN 55,212-2 micelles exhibit significantly enhanced cytotoxicity compared to free WIN in LN18 (IC50: 12.48 µM vs. 20.97 µM) and A172 (IC50: 16.72 µM vs. 30.9 µM) glioblastoma cell lines, likely due to improved cellular uptake via micellar delivery. SMA-WIN uniquely upregulates PPAR-γ (2.3-fold) in mesenchymal A172 cells, a novel finding absent in free WIN treatment, while both formulations robustly modulate CB1 and CB2 expression in epithelial LN18 cells to drive apoptosis. These subtype-specific effects underscore SMA-WIN’s potential as a targeted therapy for glioblastoma, particularly for resistant mesenchymal subtypes, addressing critical barriers in drug delivery and solubility. However, the in vitro nature of this study limits insights into tumor microenvironment interactions, necessitating further research to validate these findings in physiological contexts.

## Figures and Tables

**Figure 1 ijms-26-04544-f001:**
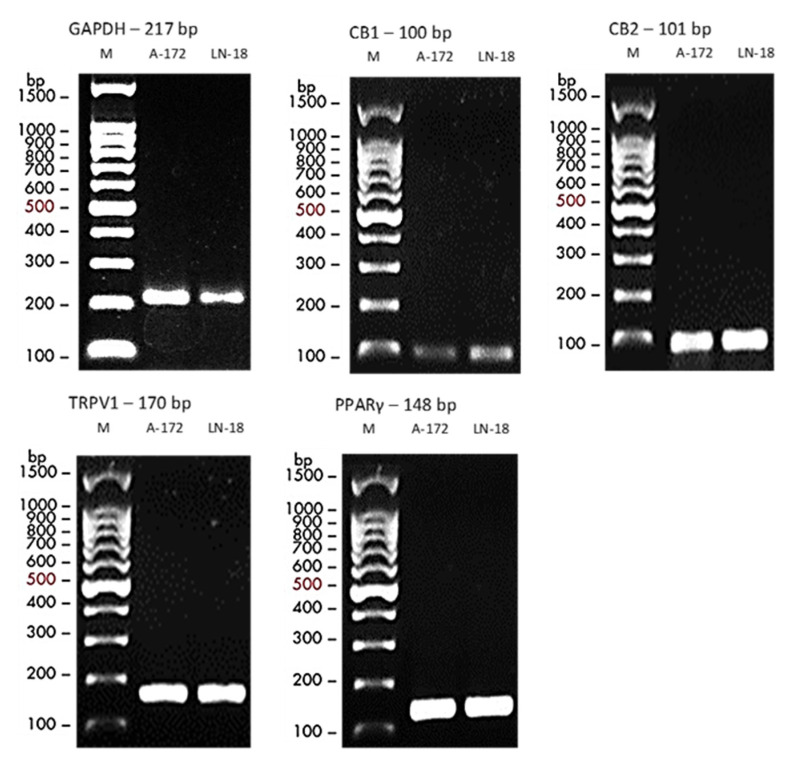
Baseline expression of cannabinoid receptors in LN18 and A172 cell lines. RT-PCR products for CB1, CB2, TRPV1, PPAR-γ, and GAPDH (housekeeping gene) were visualized on 2% agarose gels. Both cell lines showed detectable receptor expression, confirming a functional cannabinoid signaling system.

**Figure 2 ijms-26-04544-f002:**
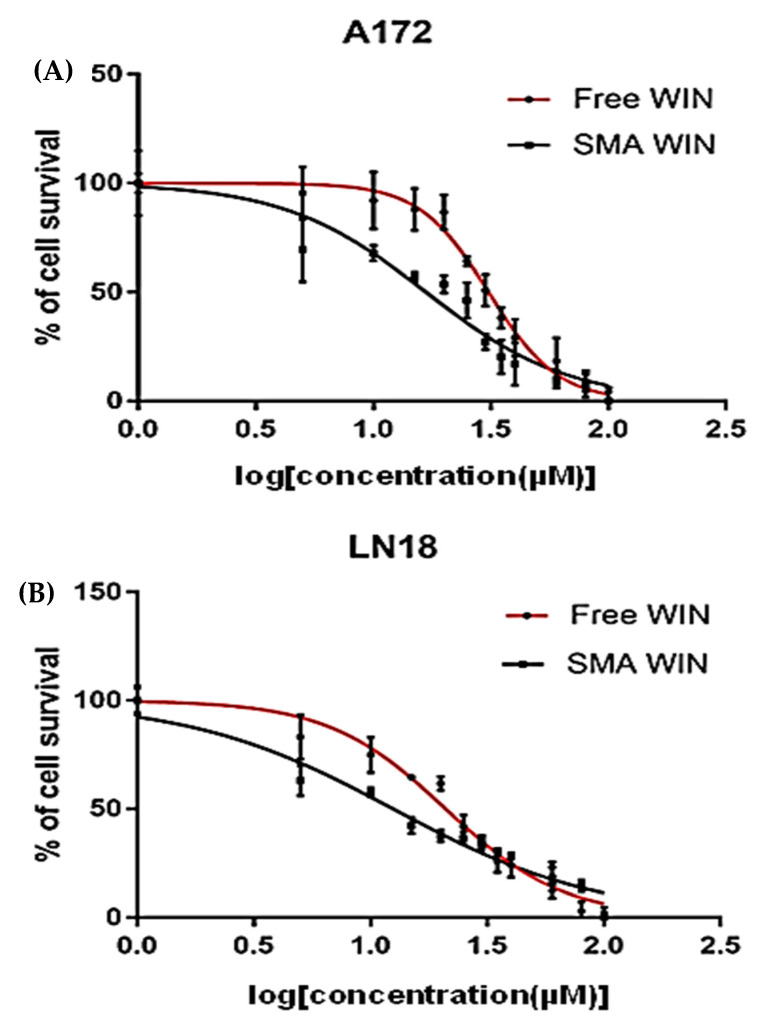
Cytotoxicity of free WIN and SMA-WIN in GBM cell lines. Dose–response curves for (**A**) LN18 and (**B**) A172 cells treated with 5–100 µM free WIN (red) or SMA-WIN (black) for 48 h, assessed by SRB assay. Data show means ± SEM from three experiments, with IC50 values indicating SMA-WIN’s greater potency (LN18: 12.48 vs. 20.97 µM; A172: 16.72 vs. 30.9 µM, *p* < 0.001). Concentrations are plotted on a logarithmic scale (µM). Detailed raw cytotoxicity data, including cell survival percentages at each concentration, are provided in Appendix A.

**Figure 3 ijms-26-04544-f003:**
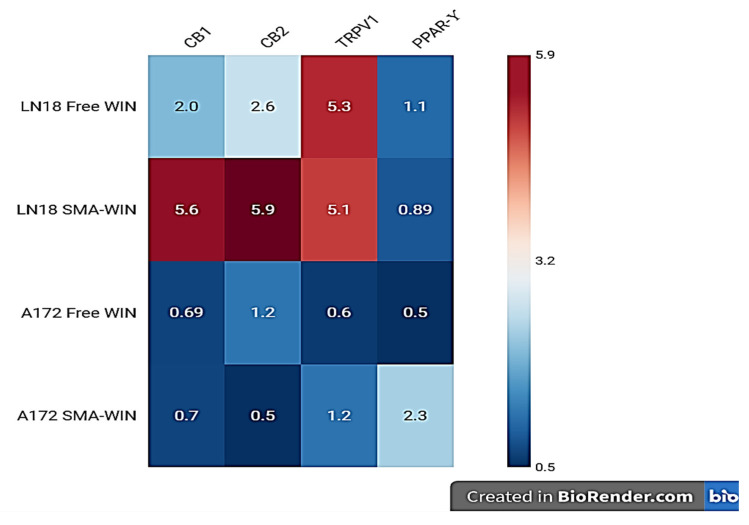
Heatmap of receptor expression changes in LN18 and A172 glioblastoma cells. Fold changes (FCs) for CB1, CB2, TRPV1, and PPAR-γ after 48 h treatment with free WIN 55,212-2 (free WIN) or SMA-WIN 55,212-2 (SMA-WIN) are shown. Red indicates significant upregulation (FC > 1.5, *p* < 0.05), blue indicates significant downregulation (FC < 0.67, *p* < 0.05), and white indicates non-significant changes (FC between 0.67 and 1.5, or *p* ≥ 0.05). Data are derived from RT-PCR analysis, with fold changes relative to untreated controls. SMA-WIN’s notable PPAR-γ upregulation (2.3-fold) in mesenchymal A172 cells highlights its subtype-specific mechanism.

**Table 1 ijms-26-04544-t001:** Primers used in real-time RT-PCR experiments.

Gene	Forward Primer (5′-3′)	Reverse Primer (5′-3′)	Amplicon Size (bp)
CB1	AAGACCCTCATCACCATCCT	GTTGATGAGGCCTTCGGGAA	100
CB2	CGGAGCTCATGCTGTCTTTA	TCAGGAAGGTCCAGGTCATC	101
TRPV1	AGCCATTGAGCATGGCATAG	GTGATGTCCTTGGTGTCCAG	170
PPAR-γ	GAGATCACAGAGTATGCCAA	CTGTCATCTAATTCCAGTGC	148
GAPDH	GGAGCGAGATCCCTCCAAAA	GGCTGTTGTCATACTTCTCA	218

**Table 2 ijms-26-04544-t002:** Cytotoxicity of free WIN 55,212-2 (free WIN) and styrene–maleic acid WIN 55,212-2 (SMA-WIN) on glioblastoma cell lines (LN18 and A172).

Cell Line	Maximum Inhibition (% ± SEM)		*p*-Value	IC50 (µM ± SEM)		*p*-Value
	Free WIN	SMA-WIN		Free WIN	SMA-WIN	
LN18	71.62 ± 0.032	80.49 ± 0.018	0.03	20.97 ± 0.080	12.48 ± 0.11	4.40 × 10^−8^
A172	54.17 ± 0.058	58.29 ± 0.041	0.05	30.9 ± 0.120	16.72 ± 0.098	8.94 × 10^−9^

Note: Maximum inhibition and IC50 values were determined using the SRB assay after 48 h treatment, with concentrations ranging from 5 to 100 µM, consistent with prior studies focused on key endpoints (maximum inhibition, IC50, *p*-values), complementing Table 3’s raw data. The full Appendix A remains in the Appendix A for comprehensive reference.

**Table 3 ijms-26-04544-t003:** Summary of cytotoxicity data for free WIN 55,212-2 (free WIN) and styrene–maleic acid WIN 55,212-2 (SMA-WIN) in glioblastoma cell lines (LN18 and A172).

Cell Line	Treatment	Cell Survival at 20 µM (% ± SEM)	Cell Survival at 100 µM (% ± SEM)	IC50 (µM ± SEM)
LN18	Free WIN	52.0 ± 1.5	10.0 ± 0.8	20.97 ± 0.08
	SMA-WIN	30.0 ± 1.3	5.0 ± 0.5	12.48 ± 0.11
A172	Free WIN	60.0 ± 1.7	12.0 ± 0.9	30.9 ± 0.12
	SMA-WIN	35.0 ± 1.5	6.0 ± 0.6	16.72 ± 0.09

Note: Cell survival (%) was measured using the SRB assay after 48 h of treatment, relative to vehicle-treated controls (100%). Data represent mean ± SEM from three independent experiments. IC50 values were calculated using GraphPad Prism 7 via nonlinear regression. A full dataset, including additional concentrations, is available in Appendix A.

**Table 4 ijms-26-04544-t004:** Fold changes in receptor expression in LN18 and A172 cells treated with free WIN or SMA-WIN. Significant changes (FC > 1.5, *p* < 0.05) are shown relative to untreated controls.

Receptor	LN-18				A-172			
	Free WIN		SMA-WIN		Free WIN		SMA-WIN	
	FC	*p*	FC	*p*	FC	*p*	FC	*p*
CB1	2.01 ↑	0.0001 ***	2.60 ↑	0.0001 ***	1.10 ↑	0.0039 **	5.30 ↑	0.040 *
CB2	2.6 ↑	0.00002 ***	5.90 ↑	0.00003 ***	0.89 ↓	0.0002 ***	5.10 ↑	0.083 ns
TRPV1	0.69 ↓	0.0014 **	1.22 ↑	0.0207 *	0.60 ↓	0.003 **	0.50 ↓	0.001 ***
PPARγ	0.70 ↓	0.001 ***	0.50 ↓	0.0003 ***	1.20 ↑	0.020 *	2.30 ↑	0.001 ***

Note: Fold changes (FCs) are relative to untreated controls, with significance defined as FC > 1.5 or <−1.5 and *p* < 0.05. *p*-values: * *p* < 0.05, ** *p* < 0.01, *** *p* < 0.001; ns = not significant. ↑: Increased and ↓: Decreased.

**Table 5 ijms-26-04544-t005:** Micelle characterization.

Parameter	Value	Method
Size (nm)	120 ± 20	Dynamic Light Scattering
Zeta Potential (mV)	−30 ± 5	Electrophoretic Mobility
Drug Loading (%)	90 ± 3	UV-Vis at 320 nm
Polydispersity Index	0.2 ± 0.05	Dynamic Light Scattering

Note: Measurements from three batches.

## Data Availability

Data will be available upon request.

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
