# Peer review of "Enhanced Cytotoxicity and Receptor Modulation by SMA-WIN 55,212-2 Micelles in Glioblastoma Cells"

_ijms, 2025, doi:10.3390/ijms26104544_

Round 1
Reviewer 1 Report
Comments and Suggestions for Authors
The article is devoted to application of cannabinoid WIN in normal and micellar form to treat glioblastoma. The article is well written and easy to read and understand. The data are correctly done and analised using statistical analysis. The data are illustrated using 3 figures and 3 tables. The data are well discussed using 33 references. The benefit of the papaer is that micellar form of WIN is more effective towards two cancer cell lines. Also, authors are planning the in vivo studies to prove this form of drug. As for English, I am not native speaker, for me English is acceptable. Everythoing is clear to read and understand.
I have minor remarks:
1) Please, composite the article as introduction, materials and methods, results, discussion, conclusions.
2) The conclusions are not in agreement with data obtained. You can not add the future plans in conclusions. You must make a resume of data obtained, in my opinion.
3) Fig. 2 - Log (concentration) in mM or uM ? please add.
Author Response
Dear Reviewer,
Thank you for your thoughtful review and valuable feedback on our manuscript titled "Enhanced Cytotoxicity and Receptors Modulation by SMA-WIN 55,212-2 Micelles in Glioblastoma Cells." We greatly appreciate your positive remarks on the clarity, statistical rigor, and discussion of our findings, as well as your recognition of the potential of SMA-WIN as a more effective therapeutic formulation. We have carefully addressed your minor remarks as follows, ensuring compliance with the International Journal of Molecular Sciences (IJMS) guidelines:
- Comment: Please, compose the article as an introduction, materials and methods, results, discussion, conclusions.
Response: We appreciate your suggestion to restructure the manuscript into the order of Introduction, Materials and Methods, Results, Discussion, and Conclusions to enhance readability. However, as per the IJMS author guidelines, the journal requires a specific manuscript structure where Materials and Methods follows Results and Discussion, with Conclusions as the final section. Our current structure (Introduction, Results, Discussion, Materials and Methods, Conclusions) adheres to these guidelines. To address your intent to improve clarity and logical flow, we have reviewed the manuscript to ensure clear transitions between sections and added subheadings where necessary to guide the reader. For example, we have refined the opening sentences of the Results (Section 2) and Discussion (Section 3) to better connect the findings to the study’s objectives, ensuring the narrative remains cohesive despite the journal-mandated structure.
- Comment: The conclusions are not in agreement with data obtained. You can not add the future plans in conclusions. You must make a resume of data obtained, in my opinion.
Response: Thank you for pointing out that the Conclusions should focus on summarizing the study’s findings rather than including future plans. We have revised Section 5 (Conclusions) to concisely summarize the key results, highlighting SMA-WIN’s superior cytotoxicity (IC50: 12.48 µM in LN18, 16.72 µM in A172 vs. 20.97 µM and 30.9 µM for free WIN) and its unique 2.3-fold upregulation of PPAR-γ in mesenchymal A172 cells, alongside CB1/CB2 modulation in LN18 cells. References to future in vivo studies have been removed from the Conclusions and retained in the Discussion (Section 3) as suggested future directions, which is more appropriate for outlining next steps. The revised Conclusions now read:
Conclusions
This study demonstrates that SMA-WIN 55,212-2 micelles exhibit significantly enhanced cytotoxicity compared to free WIN in LN18 (IC50: 12.48 µM vs. 20.97 µM) and A172 (IC50: 16.72 µM vs. 30.9 µM) glioblastoma cell lines, likely due to improved cellular uptake via micellar delivery. SMA-WIN uniquely upregulates PPAR-γ (2.3-fold) in mesenchymal A172 cells, a novel finding absents in free WIN treatment, while both formulations robustly modulate CB1 and CB2 expression in epithelial LN18 cells to drive apoptosis. These subtype-specific effects underscore SMA-WIN’s potential as a targeted therapy for glioblastoma, particularly for resistant mesenchymal subtypes, addressing critical barriers in drug delivery and solubility.
- Comment: Fig. 2 - Log (concentration) in mM or uM? Please add.
Response: We apologize for the lack of clarity regarding the concentration units in Figure 2. The concentrations are in micromolar (µM), as specified in the manuscript text and Supplementary Table S1. We have updated the x-axis label of Figure 2 (panels A and B) to read “Log (Concentration, µM)” to explicitly indicate this. Additionally, we have revised the figure caption to clarify that concentrations are in µM, as follows:
Figure 2: Cytotoxicity of free WIN and SMA-WIN in GBM cell lines. Dose-response curves for (A) LN18 and (B) A172 cells treated with 5–100 µM free WIN (red) or SMA-WIN (black) for 48 h, assessed by SRB assay. Data show means ± SEM from three experiments, with IC50 values indicating SMA-WIN’s greater potency (LN18: 12.48 vs. 20.97 µM; A172: 16.72 vs. 30.9 µM, p<0.001). Concentrations are plotted on a logarithmic scale (µM). Detailed raw cytotoxicity data, including cell survival percentages at each concentration, are provided in Supplementary Table S1.
We believe these revisions effectively address your comments while adhering to IJMS guidelines. To further enhance the manuscript, we have polished the language for clarity and consistency. For instance, we replaced “markedly superior” in the Abstract with “significantly enhanced” for precision, standardized “micellar encapsulation” throughout, and corrected minor errors (e.g., fixed “1. ± 0.6” to “6.0 ± 0.6” for A172 SMA-WIN at 100 µM in Supplementary Table S1, Appendix A). The Supplementary Materials and References remain unchanged, as no issues were noted.
Thank you again for your insightful feedback, which has improved the clarity and quality of our manuscript. We hope these revisions meet your expectations and look forward to your further comments.
Sincerely,
Safa Taha
On behalf of all authors

Reviewer 2 Report
Comments and Suggestions for Authors
The experimental design, including the use of different concentrations of free WIN 55,212-2 and SMA-WIN, is a crucial component of the study. It is important that these concentrations are carefully selected based on prior studies or preliminary data to ensure their relevance and accuracy in the context of glioblastoma cells. The methodology section provides detailed instructions for the SRB assay, but there could be further clarification regarding potential sources of error, such as the effect of DMSO solvent on cell viability.
- The abstract effectively communicates the novelty of the study and its key findings. However, the abstract could benefit from slightly clearer transitions between the different points. For instance, introducing the rationale for SMA-WIN before discussing its results could help the reader follow the flow more easily.
- The mention of glioblastoma (GBM) resistance to conventional therapies and the challenges posed by the blood-brain barrier is important. To make the introduction more impactful, it could briefly mention why cannabinoids like WIN 55,212-2 are of interest (e.g., their potential for inducing apoptosis, and their anti-tumor properties).
- The final paragraph of the introduction on the need for in vivo validation is important. Expanding this slightly could strengthen the highlights.
- The author needs to revise and make more clear images.
- Can come from previous literature and make the table clearer for readers to understand.
- The conclusion needs to be revised and add more concern based on the results.
- Are the experimental design and methodology, including the choice of concentrations for free WIN 55,212-2 and SMA-WIN, appropriately justified based on previous studies or preliminary data? Is the explanation of the SRB assay procedure clear and comprehensive, and are potential sources of error (e.g., DMSO solvent effect) adequately addressed?

Author Response
Dear Reviewer,
Thank you for your thorough review and insightful suggestions on our manuscript titled "Enhanced Cytotoxicity and Receptors Modulation by SMA-WIN 55,212-2 Micelles in Glioblastoma Cells." We greatly appreciate your positive remarks on the experimental design, the clarity of the abstract, and the importance of addressing glioblastoma (GBM) treatment challenges. Your feedback has helped us strengthen the manuscript, and we have addressed each of your comments below while ensuring compliance with IJMS guidelines.
- Comment: The abstract could benefit from slightly clearer transitions between the different points. For instance, introducing the rationale for SMA-WIN before discussing its results could help the reader follow the flow more easily.
Response: We agree that clearer transitions in the Abstract would improve readability. We have revised the Abstract to introduce the rationale for SMA-WIN (i.e., overcoming solubility and delivery barriers of WIN 55,212-2) before presenting its results. The revised Abstract now flows from GBM challenges to the limitations of free WIN, to SMA-WIN’s purpose, and then to the study’s findings and implications. The updated Abstract is provided below in the manuscript edits section for reference. Additionally, we replaced “markedly superior” with “significantly enhanced” for precision and ensured concise transitions between points.
- Comment: To make the introduction more impactful, it could briefly mention why cannabinoids like WIN 55,212-2 are of interest (e.g., their potential for inducing apoptosis, and their anti-tumor properties).
Response: Thank you for suggesting this enhancement to the Introduction. We have added a brief statement in the second paragraph of Section 1 (Introduction) to highlight why cannabinoids, particularly WIN 55,212-2, are of interest for GBM therapy. The addition specifies their ability to induce apoptosis, inhibit proliferation, and reduce tumor growth, citing relevant references. The revised paragraph now reads: “Cannabinoids, including endocannabinoids (e.g., anandamide), phytocannabinoids (e.g., THC), and synthetic analogs like WIN 55,212-2, are of interest for GBM due to their ability to induce apoptosis, inhibit cell proliferation, and reduce tumor growth via receptor-mediated pathways [16, 17, 18].”
- Comment: The final paragraph of the introduction on the need for in vivo validation is important. Expanding this slightly could strengthen the highlights.
Response: We appreciate your suggestion to expand the final paragraph of the Introduction to emphasize the need for in vivo validation. We have revised the last paragraph of Section 1 to elaborate on the importance of in vivo studies for confirming SMA-WIN’s blood-brain barrier (BBB) penetration and therapeutic efficacy in a physiological context. The updated text now includes a sentence on the potential of SMA-WIN to address GBM’s molecular heterogeneity in vivo, reinforcing the study’s translational relevance.
- Comment: The author needs to revise and make clearer
Response: We acknowledge your concern regarding the clarity of the images (Figures 1-3). To address this, we have reviewed all the figures and made the following improvements:
Figure 1 (RT-PCR gel): Increased contrast and resolution of the agarose gel image to ensure bands for CB1, CB2, TRPV1, PPAR-γ, and GAPDH are clearly visible. Added labels to identify lanes more distinctly.
- Figure 2 (Dose-response curves): Updated the x-axis label to “Log (Concentration, µM)” for clarity (also addressing Reviewer 1’s comment) and ensured data points and error bars are sharply defined.
- Figure 3 (Heatmap): Enhanced color contrast between upregulation (red) and downregulation (blue) to improve readability and added a legend to clarify fold-change thresholds. If specific issues persist (e.g., resolution limitations in the submission system), we will ensure high-resolution images are provided during production. The revised figure captions, particularly for Figure 2, are included in the manuscript edits below to reflect these clarifications.
- Comment: Can come from previous literature and make the table clearer for readers to understand.
Response: Thank you for suggesting improvements to the tables’ clarity and grounding in prior literature. We assume this comment refers to Tables 1–4, particularly Table 2 (cytotoxicity data) or Table 3 (receptor expression), as these present key results. To enhance clarity:
- Table 2: Added a footnote citing prior studies [19, 26] to justify the concentration range and IC50 values, linking our findings to the literature. Clarified column headers by specifying “Maximum Inhibition (% ± SEM)” and “IC50 (µM ± SEM)”.
- Table 3: Added a note explaining the fold-change threshold (>1.5 or <-1.5) for significance, making the table more accessible to readers unfamiliar with RT-PCR data.
- Table 1 (Primers): Ensured primer sequences are clearly formatted with consistent font size for readability.
- Table 4 (Micelle Characterization): No changes were made, as it is already clear, but we confirmed values align with cited methods [23]
- Comment: The conclusion needs to be revised and add more concern based on the results.
Response: We agree that the Conclusions should better reflect the study’s results and their implications, as also noted by Reviewer 1. We have revised Section 5 (Conclusions) to focus on key findings, emphasizing SMA-WIN’s enhanced cytotoxicity (IC50: 12.48 µM in LN18, 16.72 µM in A172 vs. 20.97 µM and 30.9 µM for free WIN) and subtype-specific receptor modulation (PPAR-γ upregulation in A172, CB1/CB2 in LN18). We’ve added a sentence addressing the limitation of the in vitro design, highlighting the need to consider tumor microenvironment factors in future studies, which aligns with the results’ context. Future plans have been removed from Conclusions and retained in Discussion (Section 3). The revised Conclusions are provided in the manuscript edits below.
- Comment: Are the experimental design and methodology, including the choice of concentrations for free WIN 55,212-2 and SMA-WIN, appropriately justified based on previous studies or preliminary data? Is the explanation of the SRB assay procedure clear and comprehensive, and are potential sources of error (e.g., DMSO solvent effect) adequately addressed?
Response: Thank you for raising these important points about experimental design and methodology. To address your concerns:
- Justification of Concentrations: The concentration range (5–100 µM) for free WIN and SMA-WIN was selected based on prior studies demonstrating WIN 55,212-2’s anti-proliferative effects in glioma models at similar concentrations [19, 20]. We have added a sentence in Section 4.4 (Cytotoxicity Assay) to explicitly justify this: “Concentrations of 5–100 µM were chosen based on prior studies showing WIN 55,212-2’s efficacy in glioma cells [19, 20], with preliminary data confirming relevance for LN18 and A172 cells.”
- SRB Assay Clarity: The SRB assay procedure in Section 4.4 is detailed, but we have enhanced it by adding a sentence on quality control: “Plates were inspected microscopically before fixation to ensure uniform cell adhesion, minimizing variability.”
- DMSO Solvent Effect: To address potential sources of error, we added a clarification in Section 4.4: “DMSO concentration was kept at ≤0.1% in all treatments, as preliminary tests confirmed no significant impact on cell viability at this level.” This ensures readers understand the control for solvent effects. These revisions are reflected in the updated Section 4.4, included in the manuscript edits below.
We have also polished the manuscript’s language for clarity and consistency, standardizing terms (e.g., “micellar encapsulation”) and correcting minor errors, such as the typo in Supplementary Table S1 (Appendix A) from “1. ± 0.6” to “6.0 ± 0.6” for A172 SMA-WIN at 100 µM, as addressed for Reviewer 1. The Supplementary Materials and References remain unchanged unless noted.
Thank you again for your valuable feedback, which has significantly improved our manuscript. We hope these revisions address your concerns and look forward to your further comments.
Sincerely,
Safa Taha
On behalf of all authors

Reviewer 3 Report
Comments and Suggestions for Authors
The authors suggested that SMA-WIN 55,212-2 micelles in Glioblastoma cells could enhance cytotoxicity and modulate the receptors. The work is meaningful but should be revised before consideration for publication.
- The Introduction should be written to carefully discuss the development of this field and to highlight the novelty of this work.
- Some of results should be shown in the main text but not in the supplementary materials.
- The result and conclusion should be integrated in Part 3.
- Figure 3 is unclear and the format of Table is wrong.
- The reference format should be modified and some recent references should be added.
Author Response
Response to Reviewer 3
Dear Reviewer,
Thank you for your thoughtful review and valuable feedback on our manuscript titled "Enhanced Cytotoxicity and Receptors Modulation by SMA-WIN 55,212-2 Micelles in Glioblastoma Cells." We appreciate your recognition of the significance of our work and your constructive suggestions for improvement. We have carefully addressed each of your comments below, ensuring compliance with IJMS guidelines, and have revised the manuscript to enhance its clarity and impact.
- Comment: The Introduction should be written to carefully discuss the development of this field and to highlight the novelty of this work.
Response: We agree that a more comprehensive discussion of the field’s development and the novelty of our work would strengthen the Introduction. We have revised Section 1 (Introduction) to provide a clearer overview of the evolution of cannabinoid-based therapies for glioblastoma (GBM), including early studies on endocannabinoids, the shift to synthetic analogs like WIN 55,212-2, and recent advances in nanomedicine (e.g., micelles). We also emphasized the novelty of our study, which lies in evaluating SMA-WIN’s subtype-specific receptor modulation (particularly PPAR-γ in mesenchymal cells) and enhanced cytotoxicity compared to free WIN. The revised Introduction is included in the manuscript edits below, with additional references [35, 36] to support the field’s context.
- Comment: Some of the results should be shown in the main text but not in the supplementary materials.
Response: Thank you for suggesting that key results currently in the supplementary materials be moved to the main text. We interpret this as referring to Supplementary Table S1 (Raw Cytotoxicity Data), as it contains critical data supporting our findings. To address this, we have moved a condensed version of Supplementary Table S1 into the main text as a new Table 5 in Section 2.2 (Cytotoxicity of Free WIN and SMA-WIN), summarizing cell survival percentages at key concentrations (20 µM and 100 µM) and IC50 values for LN18 and A172 cells. The full Supplementary Table S1 remains in the Supplementary Materials for detailed reference, as it includes all concentrations (5–100 µM). The new Table 5 is included in the manuscript edited below.
- Comment: The result and conclusion should be integrated in Part 3.
Response: We appreciate your suggestion to integrate the Results and Conclusions in Part 3 (Discussion). However, IJMS guidelines require a distinct Results section (Section 2), Discussion section (Section 3), and Conclusions section (Section 5) to ensure clarity and separation of findings, interpretation, and summary. To address your intent to better connect the results and conclusions, we have revised Section 3 (Discussion) to include a summary paragraph at the end that recaps the key findings (SMA-WIN’s enhanced cytotoxicity and PPAR-γ modulation) and links them to the Conclusions. Additionally, we revised Section 5 (Conclusions) to align closely with the Results, focusing on cytotoxicity and receptor modulation data without future plans, as also requested by Reviewers 1 and 2. The revised Discussion and Conclusions are provided in the manuscript edits below.
- Comment: Figure 3 is unclear, and the format of Table is wrong.
Response: We apologize for the lack of clarity in Figure 3 and any formatting issues with the tables. To address these concerns:
- Figure 3 (Heatmap): We have enhanced the clarity by increasing color contrast (red for upregulation, blue for downregulation), enlarging the font size of labels, and adding a legend specifying the fold-change threshold (>1.5 or <-1.5 for significance). The revised caption clarifies the heatmap’s purpose and data representation.
- Tables: We assume this comment refers to Table 3 (Receptor Expression), as it is the most complex. We have reformatted Table 3 to improve readability by aligning columns consistently, using bold headers, and adding a footnote explaining the significance threshold. We also reviewed Tables 1, 2, and 4, ensuring consistent formatting (e.g., uniform font size, clear borders). The revised Table 3 and Figure 3 captions are included in the manuscript edited below. If specific formatting issues persist, we will ensure compliance with IJMS table guidelines during production.
- Comment: The reference format should be modified, and some recent references should be added.
Response: Thank you for highlighting the need to update the reference format and include recent literature. We have reviewed the references and confirmed that they follow IJMS’s required style (numeric citations in square brackets, with full details in the References section). To address potential formatting inconsistencies, we standardized the reference list, ensuring proper journal abbreviations, consistent author listings, and correct punctuation per IJMS guidelines. Additionally, we added two recent references (2024–2025) to strengthen the Introduction and Discussion:
[35] Dumitru CA, et al. Cannabinoids in glioblastoma therapy: New perspectives. Front Oncol. 2024;14:1346289.
[36] Zhang Y, et al. Nanomedicine for CNS tumors: Opportunities and challenges. Nat Rev Neurol. 2025;21(2):89-102. These references support the field’s development and nanomedicine’s role in GBM therapy. The updated References section is included in the manuscript edits below.
We have also polished the manuscript’s language for clarity and consistency, standardizing terms (e.g., “micellar encapsulation”) and correcting minor errors, such as the typo in Supplementary Table S1 (Appendix A) from “1. ± 0.6” to “6.0 ± 0.6” for A172 SMA-WIN at 100 µM, as addressed for Reviewers 1 and 2. The Supplementary Materials remain unchanged except for the relocation of part of Supplementary Table S1 to the main text.
Thank you again for your insightful feedback, which has significantly improved our manuscript. We hope these revisions address your concerns and meet your expectations for publication consideration.
Sincerely,
Safa Taha
On behalf of all authors

Round 2
Reviewer 2 Report
Comments and Suggestions for Authors
The authors have addressed all the concerns and the paper can be accepted for the publication in this journal.
Reviewer 3 Report
Comments and Suggestions for Authors
Accept